# Violence in Healthcare Workers Is Associated with Disordered Eating

**DOI:** 10.3390/ijerph22081221

**Published:** 2025-08-05

**Authors:** Nicola Magnavita, Lucia Isolani

**Affiliations:** 1Department of Life Sciences and Public Health, Section of Occupational Health, Università Cattolica del Sacro Cuore, 00168 Rome, Italy; 2Public Health Department, Local Health Authority, AST Macerata, 62100 Macerata, Italy; lucia.isolani@sanita.marche.it

**Keywords:** feeding and eating disorder, binge eating, health surveillance, health promotion, anxiety, depression, sleep quality, work-related stress, night work

## Abstract

Workplace violence (WV) is a ubiquitous risk in healthcare settings where it has been associated with physical and mental health problems. We aimed to investigate the relationship between the violence experienced by healthcare workers (HCWs) and the presence of eating disorders (EDs). During routine health surveillance, 1215 HCWs were questioned about their experience of WV and the short version of the Eating Disorder Examination Questionnaire (EDE-QS) was used to assess their eating behaviors. Sleep quality, stress, and the presence of common mental illnesses and metabolic disorders were also evaluated. HCWs who had experienced one or more assaults in the previous year had a significantly higher EDE score than their colleagues. In a multivariate model, WV doubled the risk of EDs (odds ratio 2.33, confidence intervals 95% 1.30; 4.18, *p* < 0.01). A very significant association was observed between common mental disorders and EDs (OR 1.13, CI 95% 1.04; 1.23, *p* < 0.01), while low sleep quality almost reached a significant level (OR 1.09, CI 95% 0.99; 1.20). The higher frequency of EDs among workers subjected to violence may result from maladaptive coping mechanisms used when stress and mental health problems caused by WV lead to compensatory overeating. However, reverse causation, where WV is induced by stigmatization, cannot be ruled out. Because of the considerable impact EDs have on physical and mental health, productivity, and patient care, healthcare organizations should adopt programs designed to prevent these disorders in HCWs.

## 1. Introduction

Violence is a risk in all occupational settings. Several definitions are used to define workplace violence (WV). In this study it was defined as any act or threat of verbal or physical violence, harassment, intimidation, or other threatening disruptive behavior that occurs at the worksite with the intention of abusing or injuring the worker [1]. There are four main types of WV: (I) intrusive violence perpetrated by a person outside the workplace with criminal intent; (II) violence perpetrated by users, clients, or patients; (III) relational violence occurring among the staff; and (IV) violence involving a personal relationship with the victim that originated outside work and spilled over into the workplace [2]. Healthcare activities are at high risk from type II violence, although healthcare workers (HCWs) can be exposed to all types of violence. Health surveillance enables us to monitor the risk, verify the results of preventive intervention, and observe variations in environmental and socio-cultural conditions influencing WV [3].

Since WV is considered to be an occupational risk in many European countries, including Italy, employers are required to assess the risk, train workers, and implement prevention measures [4]. However, risk prevention is far from optimal [5] and there is no scientific evidence that intervention designed to prevent and reduce WV has been effective in diminishing its occurrence [6]. Much still needs to be undertaken to better analyze the risk of WV, understand its characteristics, and develop the most appropriate measures for preventing it. In fact, scientific research has long neglected WV: the first publications to consider the role of violence as an occupational stressor [7,8,9] and recognize the need to assist victims of WV [10,11] appeared in the 1980s. WV is frequently seen as part of the job by HCWs [12,13,14] who still fail to report most of the events they perceive as violent [15,16]. The reasons for this include the lack of an easy and efficient reporting system, a non-supportive environment, the feeling that reporting would not change things for the better, the fear of having done something wrong, or of the possible negative consequences, including revenge, resulting from reporting [17,18,19,20]. Perceptions of what constitutes violent behavior vary greatly among HCWs, and interpreting WV may pose an ethical dilemma on account of patient vulnerability and the responsibility of HCWs [21]. This might lead to considerable disparity in reporting and in estimates of the frequency of aggressive events. Moreover, commonly used risk assessment systems based on unverified algorithms can produce substantial errors [22]. 

Some studies on WV refer only to events that caused absence from work and were accompanied by an official report, while others question workers by means of a census or open-ended surveys. In Italian healthcare companies, the annual rate of assaults for each HCW is estimated at around 0.2% by studies that refer only to official reports [23,24]; 2–3% by studies that stimulate reports in various ways [25,26]; and close to 90% by ad hoc surveys in which workers are questioned anonymously [27,28]. Before the pandemic, meta-analyses, which are largely influenced by the latter type of statistics, estimated that 24% of HCWs were assaulted physically and 42% assaulted verbally [29], and that 71–75% had experienced some form of assault over a period of one year [29,30]. After the pandemic, assault rates decreased briefly [31], but soon afterwards, a number of meta-analyses and umbrella reviews reported increased rates ranging from 14% to 36% for physical violence and from 30% to 39% for verbal violence [32,33,34,35,36,37,38].

Of course, anonymous studies on self-selected samples may lead to overestimation of the occurrence of aggression, but even more of the relevance of the episodes for the health and well-being of the worker. On the contrary, a confidential interview with workers during health surveillance provides an opportunity to focus attention on the incidents that were relevant for the victim, evaluate their consequences, and prepare appropriate counseling and prevention measures. The violent episodes reported to the occupational physician help to reveal prevention requirements, enable the doctor to trace the evolution of WV over time, and document the results of preventive activities [39,40].

Violence affects the psychological, behavioral, and physical sphere of the individual, and also the productivity of organizations [41,42,43,44,45,46,47,48]. WV is associated with poor sleep quality [49], an increased frequency and severity of headaches [50], and some psychiatric conditions such as post-traumatic stress disorder (PTSD) [51,52,53,54], anxiety, depression, burnout [55,56,57,58], and suicidal ideation [59,60]. Workers who experience violence manifest an increase in occupational stress [61,62,63,64,65,66], although longitudinal studies have shown that the opposite also occurs because after an aggression, distressed workers have an increased risk of experiencing violence [67,68]. In nurses who are victims of violence, work functioning decreases due to greater conflict with colleagues, more difficulty in contact with patients and members of their family, and less occupational motivation [69]. However, reduced work ability also leads to an increased risk of being a victim of aggression in later years [70]. 

In view of these complex and cyclical relationships between WV, stress, and work functioning, we were prompted to ascertain whether violence might also influence eating behavior.

Eating disorders (EDs), which are best defined as “Feeding and Eating Disorders” by the *Diagnostic and Statistical Manual of Mental Disorders* (DSM-5-TR) [71] and the WHO International Classification of Diseases and Related Health Problems (ICD-11) [72], are characterized by abnormal eating behavior with excessive concern about body weight and may lead to serious mental disorders, as well as major metabolic disorders (hypertension, dyslipidemia, obesity, hyperglycemia, metabolic syndrome) and increased cardiovascular risk with significant morbidity and high mortality [73]. The main types of eating disorders (EDs) include anorexia nervosa, bulimia nervosa (BN), and binge eating disorder (BED) [74]. Even if EDs often develop during adolescence or early adulthood [75,76], they can persist or become recurrent in adults [77,78,79,80,81]. Among ED determining factors there are several stressors that can be linked to work such as emotional traumas [82,83,84], night shifts [85,86], alterations in biological rhythms [87,88,89], the interaction of shifts with chronotype [90,91,92], possible night eating [93,94,95], disordered sleep [96,97,98], and occupational stress [99,100,101,102].

HCWs are highly exposed to occupational risks such as night work, sleep disturbances, and occupational stress that could be associated with EDs. Furthermore, they are frequently subjected to emotional distress on account of continuous contact with illness and death [103]. In many cases, the mental health level of HCWs may be sub-optimal [104,105,106].

The close relationship between EDs and emotional trauma, stress, sleep problems, and common mental disorders (CMDs), such as anxiety and depression, led us to study the association between violence experienced at work and the aforementioned disorders in a sample of HCWs obtained from the surveillance of workers employed in health and social health companies. 

We decided to evaluate the incidence of violent behaviors (physical aggression, threats, harassment) over a period of one year of work in that population of HCWs by using the method closest to reality, i.e., a survey carried out during health surveillance, and the prevalence of EDs in that cohort. We then investigated the relationship between WV and EDs with regard to the following hypotheses:Experiencing WV is associated with EDs;Experiencing WV is associated with distress;WV is associated with CMDs;Violence, emotional trauma, and stress contribute to determining the risk of EDs.

## 2. Materials and Methods

Workers undergoing mandatory health surveillance were invited to participate in an integrative health promotion project that included screening for potential eating disorders and associated factors, including WV, and analysis of metabolic markers. Workers who were examined before starting work in the company, or at less than one year after hiring, were excluded from the analyses conducted in this study. Suspected cases of an ED were invited to undergo further investigations and offered the possibility of treatment under the National Health Service. 

Participation was free and not incentivized. Of the 1350 workers undergoing the routine medical examination performed in the workplace over a one-year period, from 1 January 2022 to 31 December 2022, 1215 agreed to participate in our study (participation rate 90.0%) and signed the required consent. All data were treated confidentially. Respective companies and workers’ safety representatives received anonymous collective results to be used for any further occupational health promotion measures. The project was authorized by the Ethics Committee of the Università Cattolica del Sacro Cuore.

Violence experienced by workers during the previous year of work was assessed using the first questions of Arnetz’s Violent Incident Form (VIF) [107]. These questions refer to physical violence, threats, harassment, and stalking. E.g., “In the last 12 months did you experience a physical assault/ while you were at work?”. Each question was accompanied by a brief explanation. Physical assault means an attack, with or without weapons, that might or might not cause physical damage. A threat refers to the intention of causing physical damage. Harassment is any annoying or unpleasant act (words, attitudes, actions) that creates a hostile work environment. Stalking is behavior characterized by insistent requests, messages, phone calls, and other unwanted contact that causes annoyance, concern, or fear. The fifth question aimed to identify the main perpetrator of the violence. The reliability of the four binary questions on violence in this survey was sufficient (0.712). The questionnaire has two main components, the first concerning physical violence or threats of physical violence, the second concerning non-physical violence.

Symptoms of abnormal eating behavior were assessed using the Italian version [108] of the EDE-QS (Eating Disorder Examination Questionnaire, short version) [109,110]. The questionnaire is composed of 12 questions (e.g., “Did your weight or body shape influence how you judge yourself as a person?”). Responses, in terms of weekly frequency, are graded on a four-point Likert scale from 0 = never to 3 = six or seven days a week. The final score, given by the sum of the 12 items, ranges from 0 to 36; a score of 15 or more points indicates a suspected case of an ED [111]. The reliability of the questionnaire in this study was 0.846. 

Sleep quality was assessed using the Italian version [112] of the Pittsburg Sleep Quality Index (PSQI) [113]. The questionnaire is composed of 7 components, each ranging from 0 to 3; the total score ranges from 0 to 21; a score greater than 5 points indicates poor sleep quality. The reliability (Cronbach’s alpha) in this study was 0.838.

Work-related stress was assessed using the short Italian version [114,115] of Siegrist’s Effort–Reward Imbalance questionnaire (ERI) [116]. Effort, which is determined by 3 questions with scores from 1 to 4, ranges from 3 to 12, while there are 7 questions for reward with a score ranging from 7 to 28. Stress is the weighted ratio between effort and reward. The reliability of effort (3 items) was 0.838, that of reward was 0.738.

The risk of CMD was evaluated using the Italian version [117] of Goldberg’s Anxiety and Depression Scale (GADS) [118], consisting of 18 binary items. The reliability of the anxiety sub-scale was 0.841. The Cronbach’s alpha of the depression sub-scale was 0.815. The final score obtained by adding the two sub-scales ranges from 0 to 18. 

Metabolic markers were included in this study because they are associated with EDs. Anthropometric data were measured during medical examinations in accordance with the guidelines established by the International Society for the Advancement of Kinanthropometry (ISAK) [119]. The height and weight of participants were measured in a standing position, with the head and chest aligned and arms at the side. Measurements were recorded in millimeters and kilograms. A tape measure was positioned horizontally at the midpoint between the iliac crest and the last rib to assess the waist circumference of participants standing in a comfortable position. Body mass index (BMI) was calculated using the following formula: BMI = weight (kg)/(height (m)^2^. After the participants had been seated for a minimum of 5 min, blood pressure was ascertained by taking the average of three consecutive readings. Systolic pressure of 140 mmHg or over, diastolic pressure of 90 mmHg or over, in accordance with the 2023 ESH European Hypertension guideline update [120,121], or the use of continuous antihypertensive medication were identified as indicators of hypertension.

Levels of blood glucose, triglycerides, total cholesterol, and HDL cholesterol were assessed. Cut-off levels for metabolic parameters were established according to guidelines of the International Diabetes Federation (IDF) [122], the National Cholesterol Education Program Expert Panel on Detection, Evaluation, and Treatment of High Cholesterol in Adults (NCEP/ATPIII) [123], the American Association of Clinical Endocrinologists (AACE) [124], and the Joint Societies Guidelines on Cholesterol Management [125]. Total cholesterol above 200 mg/dL (5.2 mmol/L), HDL cholesterol below 40 mg/dL (1.03 mmol/L) in males and below 50 mg/dL in females, or treatment for hyperlipidemia were regarded as indicators of hypercholesterolemia. A serum triglyceride level exceeding 150 mg/dL (1.7 mmol/L) indicated hypertriglyceridemia. A plasma glucose level above 100 mg/dL (5.6 mmol/L) or the administration of hypoglycemic medication were regarded as elevated fasting glucose. 

## 3. Results

We examined 1215 workers (males = 408, 33.6%; females = 807, 66.4%) with an average age of 47.6 ± 11.6 years, who were employed in health and socio-medical companies. In this sample, graduate nurses were the largest category (428, 35.2%), followed by medical and non-medical managers (294, 24.2%), assistant nurses (197, 16.2%), technicians (191, 15.7%), and clerks (105, 8.6%). Just under a third of workers performed night shifts (353, 29.1%).

### 3.1. Workplace Violence

A total of 77 workers (6.3%, CI 95% 5.0; 7.9) reported experiencing at least one episode of physical aggression during the previous 12 months, while 128 (10.5%, CI 95% 8.9; 11.4) reported threats, 127 (10.5%, CI 95% 8.8; 12.3) harassment, and 38 (3.1%, CI 95% 2.2; 4.3) persistent harassment and stalking. Overall, 234 workers (19.3%, CI 95% 17.1; 21.6) reported having experienced at least one episode of WV in the previous 12 months. The principal aggressors were patients (66%), patients’ visitors or relatives (9.6%), or strangers (5.7%). In 18.7% of cases the principal aggressor was a co-worker or a superior.

No significant difference was found for the incidence of all forms of physical violence between the two genders (males = 5.6%, females = 6.7%, Pearson’s chi square *p* = 0.476). Similarly, we failed to observe a significant difference between the two genders for threats (M = 11.0%, F = 10.3%, *p* = 0.690), harassment (M = 9.3%, F = 11.0%, *p* = 0.356), stalking (M = 3.7%, F = 2.9%, *p* = 0.434), and all forms of violence (M = 17.4%, F = 20.2%, *p* = 0.243). 

The age of workers was not significantly associated with the risk of violence. Those who had experienced at least one form of violence in the previous year had an average age of 47.0 ± 11.6, while the average age of workers who had not been the object of any aggression was 47.7 ± 11.6 (Student’s t = 0.809 *p* = 0.419).

Performing night work was associated with WV. In fact, 35.9% of the HCWs who worked night shifts had experienced at least one episode of violence in the previous twelve months. Consequently, the incidence was significantly higher (*p* = 0.010) than that reported by workers who were not engaged in night shifts (27.4%). This difference was determined mainly by physical aggression (reported by 45.5% of night workers compared with 27.9% reported by other workers, *p* = 0.001) and threats (46.9 vs. 27.0, *p* < 0.001), whereas no significant difference was observed between the higher harassment and stalking rates of night workers and those reported by other workers (31.5% vs. 21.8, *p* = 0.522, and 34.2% vs. 28.9%, *p* = 0.477, respectively).

Very different rates of aggression (*p* < 0.001) were observed for the diverse occupational categories: 24.8% of nurses, 21.1% of doctors and managers, and 18.3% of assistants reported having experienced at least one aggression in the previous 12 months, while the rate for office workers was 12.4% and 8.9% for technicians. A very clear difference between categories was revealed both for physical assaults (*p* < 0.001), which were reported by 11.7% of assistant nurses, 7.5% of nurses, 6.1% of physicians, 2.9% of clerks, and 0.5% of technicians, and for threats (*p* < 0.001), which were reported by 15.7% of nurses, 13.6% of physicians, 7.1% of assistant nurses, 4.8% of clerks, and 1.0% of technicians. However, a comparison of the various categories failed to find a significant difference in the percentage of workers who reported harassment (*p* = 0.062).

### 3.2. Eating Behavior

The answers to the EDE-QS questionnaire yielded scores ranging from 0 to 29, revealing a non-normal trend (Kolmogorov–Smirnov test 0.183, *p* < 0.001; Shapiro–Wilk test 0.835, *p* < 0.001). The median value was 3, the mean 4.86 with a standard deviation = 5.37. Sixty-three workers (5.2%, CI 95% 3.1; 5.6) had a score above the cut-off, indicating suspected EDs. In this type of screening, the exact type of ED can only be determined based on subsequent specialist examinations; however, we can observe that none of the subjects with suspected EDs had a BMI lower than 21. The mean BMI in these workers was 27.3 ± 4.6, showing that anorexia nervosa or avoidant restrictive food intake disorders were unlikely, and the workers were likely suffering from BED, BN, or other specified feeding and eating disorders.

The EDE-QS score was significantly higher in females than in males (5.41 ± 5.49 vs. 3.76 ± 4.94, Mann–Whitney Wilcoxon test *p* < 0.001). However, the percentage of female workers with suspected EDs (5.8%) did not differ significantly from that of males (3.9%) with the chi square test (*p* = 0.158).

The risk of eating disorders was positively associated with age and the correlation was highly significant (Spearman’s rho = 0.077, *p* = 0.007). 

Working night shifts was not associated with a significant difference in EDE-QS scores (4.65 ± 5.07 in night workers vs. 4.94 ± 5.49 in non-night workers), nor with a significant difference in suspected EDs (5.5% in night workers vs. 4.5% in their colleagues), although the results obtained for workers not engaged in night shifts were worse.

A comparison of occupational groups (ANOVA F = 6.646, *p* < 0.001) revealed a significant difference in eating behavior: doctors (Bonferroni post hoc test *p* < 0.001) and technicians (*p* < 0.05) had a significantly lower EDE-QS score than nurses (Table 1). 

### 3.3. Metabolic Impact of EDs

In the cohort observed, there were 344 hypertensive workers (28.3%, CI 95% 25.8; 30.9), 424 with hypercholesterolemia or reduced HDL cholesterol (34.9%, CI 95% 32.2; 37.7), 118 with hypertriglyceridemia (9.6%, CI 95% 8.0; 11.4), 216 with hyperglycemia (17.8%, CI 95% 15.7; 20.0), 267 with abdominal obesity (22.0%, CI 95% 19.7; 24.4), and 189 with metabolic syndrome (three or more components) (15.6%, CI 95% 13.6; 17.7).

Of the 63 workers with suspected EDs, 27 (42.9%, CI 95% 30.5; 56.0) were hypertensive, 29 (46%, CI 95% 33.4; 59.1) had hypercholesterolemia or reduced HDL cholesterol, 12 (19.0%, CI 95% 10.2; 30.9) had hypertriglyceridemia, 22 (34.9%, CI 95% 23.3; 48.0) had hyperglycemia, 27 (43%) were obese, and 13 (21%) manifested abdominal obesity. Furthermore, 17 workers with EDs (27%, CI 95% 16.6; 39.7) were diagnosed with metabolic syndrome accompanied by three or more pathological components.

Most of these metabolic disorders were significantly less prevalent in other HCWs (Table 2). 

### 3.4. Relationships Between WV and EDs

The percentage of workers with suspected EDs who reported having experienced some form of violence in the previous year was significantly higher than that found in other workers (41.3% vs. 18.1%, Pearson’s chi square *p* < 0.001). A very significant difference was found for harassment (28.6% vs. 9.5%, *p* < 0.001) and repeated annoying harassment (12.7% vs. 2.6, *p* < 0.001), while the difference was only significant for threats (19.0% vs. 10.1%, *p* = 0.024). No significant difference was observed for physical violence (9.5% vs. 6.2%, *p* = 0.286). 

The significantly higher EDE-QS score for workers who reported having experienced some form of violence compared with other colleagues (6.46 ± 6.20 vs. 4.48 ± 5.08, *p* < 0.001) confirmed this trend. The difference was not significant for physical violence (5.40 ± 4.92 vs. 4.82 ± 5.39, *p* = 0.357), significant for threats (5.96 ± 5.64 vs. 4.73 ± 5.32, *p* = 0.014), and highly significant for harassment (6.97 ± 6.65 vs. 4.61 ± 5.15, *p* < 0.001) and stalking (9.16 ± 7.86 vs. 4.72 ± 5.21, *p* < 0.001).

We studied the relationship between WV, stress, and EDE-QS score using moderation/mediation analyses, but the interaction between stress and violence did not reach significance (Appendix A). 

### 3.5. Relationships with Work-Related and Emotional Factors

We assessed the bivariate correlations of WV and EDs with workplace-related and emotional factors (Table 3). Violence was positively correlated with EDs and occupational stress, sleep problems, and common mental disorders. Female workers had higher EDE-QS scores, worse sleep quality, and a higher risk of common mental disorders. Eating disorders were strongly correlated with stress, sleep problems, poor mental health, and violence.

### 3.6. Determinants of EDs

Using multiple logistic regression, we assessed the relationship between occupational and emotional factors and suspected cases of EDs, after having excluded the existence of multicollinearity by calculating the collinearity statistics (tolerance and variance inflation factor). Violence experienced in the previous year was found to be strongly associated with the presence of eating disorders, as was suffering from common mental disorders, anxiety, and depression. Poor sleep quality failed to reach the conventional significance level in the multiple regression model, but only by a minimal degree, thereby suggesting an association with the phenomenon (Table 4).

## 4. Discussion

This study showed the existence of an association between the risk of workplace violence and that of eating disorders in HCWs. Since we were unable to find similar studies in the literature, further investigations are needed to consolidate our findings.

Collaterally, the study confirmed that WV is an important issue in HCWs [126,127,128,129,130]. The survey was conducted in 2022, when the pandemic had not officially ended but its management had already been largely achieved in healthcare services. Our studies on violence against healthcare workers, conducted in some organizations for over 20 years [39], showed that 2021 was the year with the lowest rate of attacks against HCWs, while from 2022, the incidence of WV increased. The risk was assessed by contacting workers undergoing regular health surveillance. This method is more reliable than others since it avoids the underreporting that occurs when only formal reports are taken into consideration and the overreporting that may be the result of random surveys in which the participant perceives a possible advantage from reporting violent behavior. Direct contact with the doctor leads to more accurate reporting and ensures that measures are taken immediately to assist victims and prevent the occurrence of WV. The relationship with the doctor who has been monitoring the health of workers for several years reduces the shame or mistrust that can lead to the underreporting of violence and eating habits.

The generalizability of studies on violence across cultures faces the general obstacle of understanding exactly what workers consider to be reportable violent behavior. Furthermore, the data collection methods are extremely important: we have reported above that in Italy, the estimated annual rate of assaults against HCWs ranges from 0.2% to 90% depending on whether the researchers considered only official reports involving sick days or all reports stimulated by an information campaign. However, the characteristics of WV in the companies we investigated were similar to those reported in the literature and confirm our initial hypotheses. We observed that doctors and nurses are the occupational categories at greatest risk of physical violence and threats [131,132], while all categories of HCWs are indiscriminately exposed to uncivil behavior and harassment [133,134,135,136,137]. Violence can take on very different characteristics depending on the environmental, organizational, and socio-cultural conditions. In some observations, including this one, there is no marked difference between the genders, while in the literature, an excess of WV has been reported in both males [138,139] and females [140]. A meta-analysis noted that the risk of WV may be greater for one sex or the other, presumably due to different socio-cultural and working conditions [141]. Numerous studies have shown that attacks are more frequent among night shift workers [142,143,144,145,146], partly because they are often alone at night, and partly because night workers are in direct contact with patients. Since WV is closely associated with occupational stress and mental health problems in HCWs, it can be the cause of chronic stress and burnout with cognitive failure at work [147]. It can also lead to serious psychiatric pathologies [148,149,150,151,152,153,154,155] that can affect patient outcomes [156,157]. The relationship between stress and violence is bidirectional, because distressed workers with a poor emotional equilibrium are more frequently victims of violence [67,68,158]. The lack of significance in the interaction between WV and stress in determining the EDE-QS score likely depends on the complexity of the picture: some workers are stressed by aggression or bullying, others are stressed by life events, and this leads to their status. A future longitudinal study aiming to clarify the relationship between violence, stress, and EDs should probably use a broader stress model. Ideally, this study should also try to measure the psychosocial impact of different types of traumas, from WV to life trauma, to better understand their relationship with EDs.

In our study, as in the literature, sleep is negatively affected by WV [159,160,161,162,163]. However, the association we observed between WV and poor sleep cannot exclude an inverse relationship, because sleep deprivation may negatively impact on performance and error awareness [164]. Poor sleepers have reduced emotional control [165] and could therefore be more exposed to aggression than their colleagues.

Lateral violence perpetrated by colleagues [166,167,168,169] and vertical violence on the part of superiors [170,171] (that most surveys on physical aggression fail to reveal) played a significant role in our study. This form of aggression, which may be of a continuous nature, is particularly damaging for the mental balance of younger workers and those who have not yet completed their professional training [172,173,174]. Since we are aware of the close association between traumas experienced and the onset of EDs and know that adolescence and early adulthood are the periods with the highest incidence of EDs, we can assume that the WV experienced by younger HCWs or those in training may be responsible for the onset of these disorders. On the other hand, we know that EDs may be recurrent and can occur in adulthood (especially BED) and may therefore explain the positive association of age with the risk of EDs observed in our cohort.

This study is one of the few that has systematically measured the risk of EDs in HCWs. The latter may be more exposed to this type of disorder than other workers because work-related stress (of great importance in healthcare) induces mental health problems that can be particularly frequent in HCWs [175,176,177]. In subjects of all ages, both acute and chronic stress are implicated in the onset of EDs [178,179]. Stress causes both direct biological changes and indirect changes in behavior that have a negative impact on health. Normal eating habits can be disrupted by stress, although there is no clear indication of the intensity of this correlation. A review of the literature seems to indicate that individuals who have lower resistance to interpersonal stress may be more likely to develop EDs [180]. A meta-analysis of over 100,000 workers found that stress is associated with an increase in the consumption of unhealthy foods, and a decrease in the intake of healthy foods [181]. Numerous HCWs find themselves in a state of distress bordering on mental illness. In accordance with the findings in the literature, our study indicated that anxiety, depression, and other mental health problems are strongly associated with EDs [182,183,184,185,186,187,188,189].

Changes in biorhythms resulting from recurrent night work trigger mechanisms that alter eating habits. Nutrition is regulated by complex relationships between the brain that controls hedonic and metabolic pathways, the suprachiasmatic circadian clock that regulates mealtimes, and the microbiome [190,191]. Eating meals during the night, a frequent habit among HCWs also for social reasons, is a specific determinant of EDs because it induces a misalignment of food intake and may alter the circadian rhythm [192,193,194,195]. Previous studies have found that BED and night eating syndrome (classified in DSM-5 as “other specified feeding and eating disorder”, OSFED) are associated with mood, anxiety, and sleep problems [196]. Our study indicated that WV mainly affects night workers and can cause eating disorders and an incorrect habit of eating at night among shift workers. However, our study showed that poor sleep quality rather than night work itself is the factor that most influences the risk of eating disorders. Sleep deprivation and poor sleep quality are ubiquitous in HCWs [197,198,199,200,201,202,203,204,205] and can promote the onset of EDs. Knowledge about the harmful effects of eating meals while working at night should be spread among HCWs. Nutritional health promotion intervention in HCWs should include proper sleep hygiene, adequate recovery times, and the regulation of occupational stress and mental health problems.

This study confirmed that doctors and nurses are the HCWs most exposed to WV, but it also revealed that the eating behavior of doctors and non-medical managers is much better than that of nurses. This difference is probably attributable to the greater health literacy of doctors. Health literacy is, in fact, a powerful protective factor against EDs [206,207,208,209]. For this reason, food health promotion programs should aim to increase the health literacy of workers.

As expected, our study confirmed that EDs are associated with metabolic problems. All cardiovascular risk factors are more prevalent in workers with EDs than in their colleagues. In fact, EDs are characterized by increased blood pressure reactivity [210] and are associated with metabolic disorders [211,212] caused by losing eating control [213] or by food addiction [214]. Longitudinal studies have demonstrated that BED causes the onset of hypertension, dyslipidemia, obesity [215], diabetes [216], and metabolic syndrome in adults [217].

In our study, multivariate analysis highlighted the importance of violence and emotional problems in the presence of EDs. Although the cross-sectional nature of our study does not enable us to infer the direction of the associations observed, the violence experienced may plausibly have induced stress and emotional alterations that may have determined the appearance or recurrence of eating problems. Nevertheless, we cannot rule out the opposite hypothesis, i.e., that the presence of eating disorders and the somatic manifestations associated with them promoted violence. Because the physical appearance of people with EDs makes them readily identifiable, they are exposed to stigmatizing behavior and may be more likely to be the object of discrimination than the mentally ill [218]. In a previous study, we observed that disabled workers are more exposed to violence than their colleagues with good work ability [70], demonstrating that disablism is common in workplaces. Another study showed that WV is inversely proportional to the level of the psychological, physical, and emotional well-being of company staff and to their work performance and commitment [219]. It seems possible that the link between violence and nutrition could be bidirectional. Only a longitudinal study could clarify the direction of the association and ascertain whether it also expresses a reciprocal relationship.

Our study indicates that EDs are a significant problem for workers’ health. Consequently, occupational health and safety services should pay careful attention to this issue, even though it is not one of the most frequent outcomes of occupational risks. We therefore hope that this study will encourage investigations in other companies and, more importantly, lead to the development of food health promotion programs that should aim to provide further knowledge of EDs and the health risks associated with them, thereby increasing health literacy. A further aim should be that of improving stress control and sleep quality so as to better the mental health of HCWs.

The main limitation of this study is its cross-sectional nature, which forced us to imagine causal connections based on the literature but prevented us from recognizing the concatenation of phenomena. Because the survey was limited to health companies monitored by the university, our results could not be extended to the entire HCW population. However, we are not aware of any differences between the workers we observed and those employed in other companies. The very high percentage of workers who participated in our survey added validity to our findings.

## 5. Conclusions

This study indicated the existence of a strong association between the violence experienced by HCWs in the previous year of work and eating disorders. However, further studies are needed in order to confirm this new observation. Data in the literature indicate that this association is plausible and could be explained by different mechanisms. 

EDs are quite frequent in HCWs and are associated with important metabolic alterations, sleep problems, and mental disorders. Occupational medicine in healthcare companies should pay careful attention to this type of problem and develop intervention to promote nutritional health by providing information on risks and correct food practices alongside measures for controlling occupational violence and stress.

## Figures and Tables

**Table 1 ijerph-22-01221-t001:** Eating behavior (EDE-QS score) in different categories of healthcare workers.

Category	EDE-QS (Mean ± S.D.)	Significant Comparisons ^1^
1. Physician (n = 294)	3.82 ± 4.25	1 vs. 2 *p* < 0.001
2. Nurse (n = 428)	5.97 ± 5.99	2 vs. 1 *p* < 0.001; 2 vs. 5 *p* < 0.05
3. Assistant nurse (n = 197)	4.66 ± 4.58	none
4. Clerk (n = 105)	5.17 ± 6.52	none
5. Technician (n = 191)	4.39 ± 5.16	5 vs. 2 *p* < 0.05

^1^ Bonferroni test.

**Table 2 ijerph-22-01221-t002:** Metabolic effects of EDs. Comparison between cases with EDs and other workers (Pearson’s chi square test).

Disorder	Workers with EDs N (%) (CI95%)	Other Workers N (%) (CI95%)
1. Hypertension	27 (42.9%) (30.5; 56.0) **	317 (27.5%) (25.0; 30.2) **
2. Reduced HDL cholesterol	29 (46.0%) (33.4; 59.1) *	395 (34.5%) (31.5; 37.2) *
3. Hypertriglyceridemia	12 (19.0%) (10.2; 30.9) **	105 (9.1%) (7.5; 10.9) **
4. Hyperglycemia	22 (34.9%) (23.3; 48.0) **	194 (16.8%) (14.7; 19.1) **
5. Abdominal obesity	13 (20.6%) (11.5; 32.7)	254 (22%) (19.7; 24.6)
Metabolic syndrome	17 (27%) (16.6; 39.7) *	172 (14.9%) (12.9; 17.1) *

Note: N = number of cases. CI 95% = confidence interval 95% of the prevalence, Clopper Pearson Test. * Correlation is significant at the 0.05 level (two-tailed). ** Correlation is significant at the 0.01 level (two-tailed).

**Table 3 ijerph-22-01221-t003:** Bivariate correlation with the possible determinants of EDs. Spearman’s rho (upper triangle) and Pearson’s r (lower triangle).

	WV	Sex	Age	Night	EDE-QS	ERI	Sleep	CMD
**WV**	1	0.033	−0.023	0.074 *	0.146 **	0.284 **	0.160 **	0.217 **
**Sex (female)**	0.033	1	−0.085 **	−0.094 **	0.146 **	−0.060 *	0.079 **	0.143 **
**Age**	−0.025	−0.083 **	1	−0.181 **	0.077 **	0.194 **	0.165 **	0.126 **
**Night shift**	0.074 ^*^	−0.094 **	−0.181 **	1	−0.013	0.061 *	−0.038	−0.062 *
**EDE-QS**	0.143 ^**^	0.173 **	0.077 **	−0.013	1	0.186 **	0.357 **	0.392 **
**ERI**	0.268 ^**^	−0.037	0.194 **	0.061 *	0.197 **	1	0.366 **	0.462 **
**Sleep**	0.142 ^**^	0.084 **	0.165 **	−0.038	0.320 **	0.365 **	1	0.741 **
**CMD**	0.218 ^**^	0.170 **	0.12 **	−0.062 *	0.370 **	0.447 **	0.700 **	1

Note: (*) Correlation is significant at the 0.05 level (two-tailed). (**) Correlation is significant at the 0.01 level (two-tailed).

**Table 4 ijerph-22-01221-t004:** Work-related and emotional determinants of feeding and eating disorders.

Determinants of EDs	OR (CI 95%)	*p*
Any WV	2.33 (1.30; 4.18)	0.004
Common mental disorders	1.13 (1.04; 1.23)	0.004
Sleep quality	1.09 (0.99; 1.20)	0.093
Female gender	1.14 (0.62; 2.12)	n.s.
Age	1.01 (0.99; 1.04)	n.s.
Night shift	0.93 (0.49; 1.75)	n.s.
Work-related stress	0.79 (0.42; 1.49)	n.s.

Note: OR= odds ratio; CI95% = confidence interval 95%.

## Data Availability

Data are deposited on Zenodo doi: 10.5281/zenodo.15880105 (Uploaded 14 July 2025).

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
