# Peer review of "Violence in Healthcare Workers Is Associated with Disordered Eating"

_ijerph, 2025, doi:10.3390/ijerph22081221_

Round 1
Reviewer 1 Report
Comments and Suggestions for Authors
The manuscript is well-written and provides a detailed and rigorous description of the methodology. The authors are to be commended for the clarity and thoroughness of their work.
If feasible, it would be beneficial to include reliability coefficients (e.g., Cronbach’s alpha) for the following instruments used in the study: Arnetz’s Violent Incident Form (VIF), the short version of the Eating Disorder Examination Questionnaire (EDE-QS), the Pittsburgh Sleep Quality Index (PSQI), Goldberg’s Anxiety and Depression Scale (GADS), and Siegrist’s Effort-Reward Imbalance questionnaire (ERI). This would enhance the psychometric transparency of the study.
Additionally, while the sample size is commendably large, the authors might consider including a measure of sample representativeness or a power analysis to further strengthen the methodological robustness.
Finally, I would like to point out a typographical error on line 419 and an unnecessary hyphen on line 464.
I encourage the authors to consider these points in their revision.
Author Response
Reviewer #1
The manuscript is well-written and provides a detailed and rigorous description of the methodology. The authors are to be commended for the clarity and thoroughness of their work.
Response: We thank the reviewer for his appreciation of our work and for his valuable contributions, which significantly improve the manuscript.
If feasible, it would be beneficial to include reliability coefficients (e.g., Cronbach’s alpha) for the following instruments used in the study: Arnetz’s Violent Incident Form (VIF), the short version of the Eating Disorder Examination Questionnaire (EDE-QS), the Pittsburgh Sleep Quality Index (PSQI), Goldberg’s Anxiety and Depression Scale (GADS), and Siegrist’s Effort-Reward Imbalance questionnaire (ERI). This would enhance the psychometric transparency of the study.
R.: We gladly accepted the suggestion, inserting the requested results
Additionally, while the sample size is commendably large, the authors might consider including a measure of sample representativeness or a power analysis to further strengthen the methodological robustness.
R.: We did not conduct a sample size study before undertaking this research because health promotion activities, such as health surveillance, must be offered to all workers. Our study therefore served as a census. Given that 90% of workers participated, the sample can be considered representative of the workers under surveillance. The characteristics of the individuals who participated in the survey closely match the characteristics of the overall population. No significant differences emerged between the HCWs monitored by us and other HCWs in other parts of the country.
Finally, I would like to point out a typographical error on line 419 and an unnecessary hyphen on line 464.
I encourage the authors to consider these points in their revision.
Response: We thank the reviewer for the attention with which he/she revised this study, and we welcome the suggestions he/she gave us.
Reviewer 2 Report
Comments and Suggestions for Authors
The association between workplace violence and eating disorders has not been extensively addressed in the literature. My compliments to the authors for the topic, study design, and results. A future longitudinal study would be needed to better elucidate any associations between eating disorders and other aspects of work-related stress, in relation to gender and aging.
Author Response
Reviewer #2
The association between workplace violence and eating disorders has not been extensively addressed in the literature. My compliments to the authors for the topic, study design, and results. A future longitudinal study would be needed to better elucidate any associations between eating disorders and other aspects of work-related stress, in relation to gender and aging.
Response: We thank the reviewer for the commitment with which he/she revised this work, and we hope that future studies will develop the ideas that emerge from it.
Reviewer 3 Report
Comments and Suggestions for Authors
The study "Violence in healthcare workers is associated with disordered eating" addresses a relevant and little explored topic in the literature. The work presents important contributions to the understanding of the impacts of violence at work (VT) on the health of health professionals (HCWs), especially in relation to eating disorders (ATs).
The association between TV and ATs in HCWs is innovative and clinically significant. The expressive sample (N = 1,215) and high participation rate (90%) are a strong points of the work. The use of validated instruments (EDE-QS, PSI, ERI) and multivariate analyses is appropriate. Integration was performed with the metabolic, psychological and occupational data, which was an important differential. The theoretical discussion is well founded and critically explored, especially with regard to stress, sleep, and stigmatization.
Suggestions:
Highlight that the data are self-reported and may underestimate VT or ATs (e.g.: Shame in reporting aggression or compulsion to eat).
The cross-sectional design does not allow inferring whether VT causes ATs or vice versa. Include a more detailed discussion about future longitudinal studies. Another suggestion would be to explore the possibility of mediation analysis (e.g., Stress as mediator).
I suggest an analysis between the different professions (e.g. Nurses vs. Doctors) or shifts (night vs. Day). If available, include data on previous history of eating or psychiatric disorders.
Include in the discussion whether the results can be generalized to other countries with different working cultures. I did not find the period of study in the manuscript. Was the study conducted post-pandemic? If so, how could this have influenced VT and the mental health of HCWs?
Better explain why markers such as hypertension and HDL were included (their relationship with chronic stress or TAs). It would be interesting to include a graphic conceptual model illustrating the hypotheses of association (would be clearer in the results). In Table 4, explain why "occupational stress" was not significant (may be due to collinearity with anxiety/depression?). There is little mention of how TAs in HCWs differ from the general population (e.g., Binge eating patterns on night shifts).
The article has great potential and already presents valuable contributions. The above suggestions aim to deepen statistical and contextual analysis, improve transparency on limitations, and make practical implications more actionable.
Author Response
Reviewer #3
The study "Violence in healthcare workers is associated with disordered eating" addresses a relevant and little explored topic in the literature. The work presents important contributions to the understanding of the impacts of violence at work (VT) on the health of health professionals (HCWs), especially in relation to eating disorders (ATs).
The association between TV and ATs in HCWs is innovative and clinically significant. The expressive sample (N = 1,215) and high participation rate (90%) are a strong points of the work. The use of validated instruments (EDE-QS, PSI, ERI) and multivariate analyses is appropriate. Integration was performed with the metabolic, psychological and occupational data, which was an important differential. The theoretical discussion is well founded and critically explored, especially with regard to stress, sleep, and stigmatization.
Response: We thank the reviewer for his/her careful evaluation of our work, highlighting the points that needed improvement.
Suggestions:
Highlight that the data are self-reported and may underestimate VT or ATs (e.g.: Shame in reporting aggression or compulsion to eat).
Response: The reviewer highlighted an important point in this type of investigation regarding the reliability of self-reporting. The method we used, matching self-reported questions to the doctor's visit, was designed specifically to reduce the conditions of mistrust or shame that can lead to underreporting, or the social desirability that can lead to overreporting. In the previous version, we had already written that "this method is more reliable than others since it avoids the underreporting that occurs when only formal reports are taken into consideration and the overreporting that may be the result of random surveys in which the participant perceives a possible advantage from reporting violent behavior. Direct contact with the doctor leads to more accurate reporting and ensures that measures are taken immediately to assist victims and prevent the occurrence of WV.” We now added that “the relationship with the doctor who has been monitoring the health of workers for several years reduces the shame that can lead to underreporting of violence and eating habits.”
The cross-sectional design does not allow inferring whether VT causes ATs or vice versa. Include a more detailed discussion about future longitudinal studies. Another suggestion would be to explore the possibility of mediation analysis (e.g., Stress as mediator).
Response: We studied the relationship between WV, stress, and EDE scores using moderation/mediation analyses, but the interaction between stress and violence did not reach significance (Table S1). We reported these data in the manuscript. In the Results section we added: “We studied the relationship between WV, stress, and EDE-QS score using moderation/mediation analyses, but the interaction between stress and violence did not reach significance (Table S1).” In the Discussion, we added: “The lack of significance in the interaction between WV and stress in determining the EDE-QS score likely depends on the complexity of the picture: some workers are stressed by aggression or bullying, others are stressed by life events, and this leads to their status. A future longitudinal study aiming to clarify the relationship between violence, stress, and EDs should probably use a broader stress model. Ideally, this study should also try to measure the psychosocial impact of different types of traumas, from WV to life trauma, to better understand their relationship with EDs.”
I suggest an analysis between the different professions (e.g. Nurses vs. Doctors) or shifts (night vs. Day). If available, include data on previous history of eating or psychiatric disorders.
Response: We consider the reviewer's suggestion to be very valid and open to interesting developments. We have observed that physicians and managers of other professions have a lower prevalence of EDs than nurses and other categories. We believe this may be due to a different level of health literacy, as previous studies have shown that health knowledge influences the risk of EDs. We found no differences related to night shift work, but there is a significant effect related to sleep quality. We have discussed the reasons for this apparent inconsistency and intend to conduct a study in a population not employed on night shifts to clarify the role of sleep regardless of shift work. Finally, we confirm that the interviewees' health records would allow us to verify the pre-existence of eating disorders prior to the survey; however, this data was not collected in this survey.
Include in the discussion whether the results can be generalized to other countries with different working cultures. I did not find the period of study in the manuscript. Was the study conducted post-pandemic? If so, how could this have influenced VT and the mental health of HCWs?
Response: We thank the reviewer for pointing out that we had overlooked the fact that the survey was conducted in 2022, when the pandemic had not officially ended but its management had already been largely achieved in healthcare services. Our studies on violence against healthcare workers, conducted in some organizations for over 20 years [39], have allowed us to analyze the effect the pandemic has had on rates of assault against staff. The generalizability of studies on violence across cultures faces the general obstacle of understanding exactly what workers consider to be reportable violent behavior. Furthermore, the data collection methods are extremely important: we have reported that in Italy the rate of assaults against healthcare workers ranges from 0.2% to 90% depending on whether the researchers considered only official reports involving sick days or all reports stimulated by an information campaign. We have added these considerations in the Discussion.
Better explain why markers such as hypertension and HDL were included (their relationship with chronic stress or TAs). It would be interesting to include a graphic conceptual model illustrating the hypotheses of association (would be clearer in the results). In Table 4, explain why "occupational stress" was not significant (may be due to collinearity with anxiety/depression?). There is little mention of how TAs in HCWs differ from the general population (e.g., Binge eating patterns on night shifts).
Response: We thank the reviewer for pointing out that in the introduction we had not specified that EDs are associated with metabolic problems (hypertension, dyslipidemia, obesity, hyperglycemia, and metabolic syndrome) and increased cardiovascular risk. We have added this notation. Metabolic markers were included because they are associated with EDs. Many of them are also related to stress and sleep disturbances. The result isn't surprising and confirms the associations we observed in this study. We raised the issue of possible night eating syndrome, even though this study did not have specific tools for identifying this disorder. We added to the discussion the consideration that "knowledge of how harmful it is to consume meals during night work should be spread among HCWs." Before conducting the multiple logistic regression test, we excluded the existence of multicollinearity by calculating the collinearity statistics (tolerance and variance inflation factor). We added this information to the results.
The article has great potential and already presents valuable contributions. The above suggestions aim to deepen statistical and contextual analysis, improve transparency on limitations, and make practical implications more actionable.
Response: We are indebted to the reviewer for the contribution he/she made to this work, and we hope that future studies will develop the ideas emerging from it.